# Epidemiology of Kaposi’s Sarcoma

**DOI:** 10.3390/cancers13225692

**Published:** 2021-11-14

**Authors:** Sophie Grabar, Dominique Costagliola

**Affiliations:** 1Sorbonne Université, INSERM, Institut Pierre Louis d’Épidémiologie et de Santé Publique (IPLESP) and Assistance Publique-Hôpitaux de Paris (AP-HP), Unit of Public Health, St Antoine Hospital, 75001 Paris, France; 2Institut Pierre Louis d’Épidémiologie et de Santé Publique (IPLESP), INSERM, Sorbonne Université, 75001 Paris, France; dominique.costagliola@iplesp.upmc.fr

**Keywords:** Kaposi’s sarcoma, cancer, epidemiology, human herpesvirus 8, KSHV, HIV infection

## Abstract

**Simple Summary:**

Kaposi’s sarcoma is a rare tumor caused by human herpesvirus 8 in the context of immunodeficiency, such as that induced by HIV infection or immunosuppressive therapy. In patients living with HIV (PLHIV), thanks to potent antiretroviral treatment that efficiently restores immunity and allows a better control of HIV infection, the occurrence of KS has decreased dramatically over the past 25 years. However, KS is still one the most frequent cancers in PLHIV, in particular in men who have sex with men and in sub-Saharan Africa, where it is still endemic. Even in the context of restored immunity, the risk of KS is still more than 30 times higher in PLHIV than in the general population.

**Abstract:**

Kaposi’s sarcoma is an angioproliferative tumor caused by human herpesvirus 8 in the context of immunodeficiency, such as that induced by HIV infection or immunosuppressive therapy. Its incidence has dramatically fallen in patients living with HIV (PLHIV) since the introduction of potent antiretroviral combinations 25 years ago due to the restoration of immunity and better control of HIV replication. However, KS is still one of the most frequently occurring cancers in PLHIV, in particular in men who have sex with men and in sub-Saharan Africa, where it is still endemic. Even in the context of restored immunity, the risk of KS is still more than 30 times higher in PLHIV than in the general population. Recent evidence indicates that early initiation of antiretroviral treatment, which is recommended by current guidelines, may reduce the risk of KS but it needs to be accompanied by early access to care. This review mainly focuses on the recent epidemiological features of KS in the context of HIV infection.

## 1. History of KS: A Disease Understood Step by Step

Kaposi’s sarcoma (KS) is an angioproliferative tumor that was first described in its “classic form” at the end of the XIX century by a Hungarian dermatologist, Moritz Kaposi, as a rare indolent cutaneous tumor affecting elderly individuals, mostly men from Eastern Europe and the area bordering the Mediterranean Sea. In the 1950s, a more aggressive and “endemic form” of KS was described in sub-Saharan Africa that affected young African adults and children [1]. In the 1960s, an iatrogenic form of KS was observed in kidney transplant recipients and patients receiving aggressive immunosuppressive therapies [2]. Then, in the 1980s, with the advent of the HIV/AIDS epidemic, a new form of KS, the epidemic form, was described in young homosexual men in New York and California [3,4] infected by a new virus, the human immunodeficiency virus (HIV). From that period onward, KS, which was previously rare and poorly studied, became an epidemic disease in patients with AIDS.

The causative agent of KS, human herpesvirus 8 (HHV-8), also reported as KSHV (Kaposi’s sarcoma herpes virus), was identified in 1994 in KS lesions obtained from patients with acquired immunodeficiency syndrome (AIDS) [5]. This breakthrough discovery confirmed the previous suggestions, based on epidemiological data, that an oncogenic virus was the etiological agent of KS [6,7]. HHV8 is a necessary cause but is not sufficient for the development of KS. It usually develops in the context of immune deficiency, such as that induced by HIV infection or immunosuppressive therapy.

The four forms of KS, the classic (Mediterranean), endemic (African), epidemic (HIV/AIDS-associated), and iatrogenic (transplant-related), share the same causative virus but have distinct epidemiological and clinical presentations. Some authors also hypothesize on the existence of a fifth form (see paragraph 5 below).

Here, we will mostly focus on the epidemiology of KS in patients living with HIV (PLHIV) and on recent insights concerning its evolution. Certain insights on post-transplantation and other KS are also provided.

## 2. Geographical and Population Disparities

HHV8 infection is not ubiquitous. Its prevalence is substantially higher in men who have sex with men (MSM) and in certain regions of the world, such as sub-Saharan Africa or the countries bordering the Mediterranean; whereas it has been estimated to affect < 5% of the general population of the USA and Europe [8]. Although the transmission routes of HHV8 are not fully understood, close sexual contact and saliva have been shown to explain the observed disparities across population groups [9]. Transmission through blood is also possible but to a lesser extent [10].

In sub-Saharan Africa, the prevalence of HHV8 also varies depending on the region. It reaches 50% in Uganda [11,12], where the prevalence is already high in early childhood, with little increase thereafter, whereas the prevalence in South Africa and Zimbabwe in early adulthood is lower (<15%) and then increases with age [12]. In a recent study investigating patients with tuberculosis symptoms, the authors reported a seroprevalence of 30.7% in Cape Town South Africa [13].

A recent meta-analysis [14] estimated the seroprevalence of HHV8 in MSM to be 33.0% (95% confidence interval (CI), 29.2–37.1) and the factors associated with such high prevalence in this population to be high-risk behavior.

Another meta-analysis [15] examined the association between HIV infection and the risk of HHV8 infection across the five continents and showed that HIV infection is associated with a high prevalence of HHV8, regardless of the region or population group. The association was strongest for MSM and children and weaker for heterosexual adults and intravenous drug users. The authors suggested several mechanisms to explain the positive association between HIV and HHV8: reactivation of pre-existing HHV8 in PLHIV [16], increased susceptibility to HHV8 infections in PLHIV [17], and shared transmission routes.

As a consequence of the heterogeneous distribution of HHV8, there are geographical disparities in the burden of KS. There were roughly 42,000 new KS cases and 20,000 deaths estimated in 2018 worldwide [18]. KS occurs at a globally higher frequency in men than in women [19] and in the HIV-infected population [20,21,22,23]. Among PLHIV, the incidence rate of KS is much higher for MSM than for other men or women (Figure 1). Of note, the sex difference in sub-Saharan Africa is less pronounced than in European or North-American countries [20].

The Globocan 2020 data [19], which compiled the worldwide estimates of cancer incidence and mortality, highlight these disparities across countries. Although KS was extremely rare in most regions of the world (global incidence rates per 100,000 PY being 0.5 in men and 0.3 in women), KS was one of the most frequently occurring cancers in sub-Saharan Africa with highest rates in Southern and Eastern Africa and occurred mostly in PLHIV. In 2012, in Eastern Africa, KS was the leading cause of cancer in men and the third in women (incidence rates of 15.1 per 100,000 PY in men and 7.6 in women) [24].

In sub-Saharan Africa, KS is still one of the most common cancers [25,26] and the KS burden has risen since the beginning of the HIV epidemic [24]. For example, the incidence of KS in Zimbabwe increased in both sexes: 7.6 fold in men and 23.9 fold in women between 1963–1972 and 2011–2015 [25] (see Figure 2).

Overall, these geographical differences are likely related to the combined effect of a higher prevalence of HHV8, a higher prevalence of HIV infection, and lower access to antiretroviral treatment (ART) in these regions. However, the incidence of KS has decreased in all regions of the world since the initiation of cART, as shown by a multiregional comparative study of PLHIV initiating combined ART (cART) [20]. This suggests that a similar decreasing trend in the incidence and burden of KS can also be expected in these regions with the increasing expansion of available ART.

## 3. KS in PLHIV

Since the very beginning of the HIV/AIDS epidemic and the first description of KS in a cluster of young MSM [3], KS has been associated with the HIV/AIDS epidemic. KS was considered to be an AIDS-defining disease by the Centers for Disease Control in 1982 and later [27]. Along with non-Hodgkin lymphoma and invasive cervical cancer, KS is one of the three AIDS-defining cancers of the revised AIDS classification of 1993 [28].

Prior to the introduction of potent antiretroviral treatment in 1996, which dramatically changed the course of HIV infection in PLHIV by increasing their life expectancy and reducing the incidence of opportunistic diseases, KS was among the most common AIDS-defining diseases in PLHIV. By 1989, 15% of the patients with AIDS in the US reported to the CDC had KS [7]. In France, KS was the first AIDS-defining illness for 13.9% of the patients with AIDS between 1993 and 1995 [29] and it was 17.2% in Australia [30] by 1991.

### 3.1. Incidence and Risk vs. That in the General Population

With the advent of combined antiretroviral treatment (cART) in 1996, and the resulting virological and immunological control of HIV infection, the incidence rate of KS has dramatically declined over time [20,22,31,32,33,34,35]. The relative reduction of the incidence of KS between the pre- and post-cART era has been more than 60% [22,36,37], from more than 3000 to less than 100/100,000 person years [35,38,39]. The decline was sharper during the first periods of cART availability and for MSM [31,33]. Between 1996 and 2003, the relative reduction in the ANRS CO4 FHDH cohort [31] was approximately 70% among MSM and 60% for others. The risk fell more sharply for KS with visceral involvement than for KS without visceral involvement (>50% and 30%, respectively) [31].

The most recent data from the USA indicate that the risk of KS is still decreasing [35,38]. In a comprehensive analysis of KS trends among PLHIV in America between 2008 and 2016, a significant continuous decline was reported nationally (−3.2% per year) but also showed stagnation of this decline in certain states and in younger and black PLHIV, probably explained by unequal access to HIV medical care, calling for a continuous effort towards the early diagnosis and treatment of HIV [38].

### 3.2. Immuno-Deficiency, HIV Viral Load, and CD4/CD8 Ratio

KS is highly associated with immunodeficiency in PLHIV [21,39,40,41,42]. It is also independently associated with HIV viral load (VL) [21,39,40,41,42]. In the study of Guiguet et al. of the large ANRS CO4 FHDH cohort [21], which investigated the incidence of cancer among more than 50,000 PLHIV followed in France between 1998 and 2006, the risk of KS (n = 565) steadily increased as the recent CD4 cell count decreased and recent viral replication rose (see Figure 1). Such an association with recent CD4 and VL values indicates that immunosuppression and HIV infection may be involved with late stage of KS development. The risk factors of KS have changed over time for PLHIV initiating cART, with HIV viral load becoming increasingly important [20]. Recently, the large North-American cohort collaboration [42] showed an independent association between cumulative VL and KS risk, suggesting that HIV infection may also promote early phases of KS development.

Although the risk of KS has declined over time in all transmission groups, the risk of KS is still 300–500-fold higher in PLHIV than in the general population [22,33,37,43]. Even in PLHIV with restored immunity (i.e., CD4 levels > 500/mm^3^), large cohorts have shown that the risk of KS is still high. Relative to the general population, the risk is 60-fold higher for patients with a recent CD4 level > 500/mm^3^ [41] and 35-fold higher for patients with both a CD4 level above 500/mm^3^ for two years and an HIV viral load < 500 cp/mL [22]. The risk is still over 50-fold higher for PLHIV with long-term viral suppression (i.e., >2 years) than for the uninfected population [44]. Moreover, some studies have reported KS occurring in aviremic patients [45,46,47].

Recently, Caby et al. [48] showed low CD4/CD8 ratios to be associated with an increased risk of KS in PLHIV on cART who achieved virological control in the large European collaboration cohort, especially when CD4 was ≥500/mm^3^. Indeed, the association was stronger in this population with restored immunity than in the whole population of virally suppressed PLHIV. The hazard ratio of KS ranged from 1.4 to 3.3 in PLHIV with restored immunity with a CD4/CD8 ratio between 0.8 and 0.3 relative to those with a CD4/CD8 ratio of 1, whereas the risk ranged from 1.2 to 2.6 in the whole population of virally suppressed PLHIV. This is an interesting finding, as the population of PLHIV with high CD4 levels is growing as a result of effective cART and early cART initiation.

### 3.3. Immune Reconstitution Inflammatory Syndrome (IRIS)

In the context of IRIS, a paradoxical worsening or the first appearance of opportunistic infections and other conditions has been described [49]. Several studies have shown that the incidence of KS is very high in the first 6 months among patients initiating cART and then decreases [23,50,51,52]. In the large European collaboration cohort [50], the incidence rate of KS was highest 6 months after starting cART, at 953 per 100,000 person-years (95%CI, 866–1048), and declined to 82 (68–100) after 5–8 years. Similar results of a high incidence immediately after cART initiation that then steeply declines have been observed, regardless of the region of the world [20].

### 3.4. Impact of Early cART Initiation on the Risk of KS

Following the results of the INSIGHT START clinical trial [53], immediate initiation of ART was recommended for all PLHIV, regardless of the CD4 count. The question arises as to whether earlier cART initiation reduces the risk of KS by preventing immune decline through reducing exposure to long periods of immunosuppression and inflammation/activation. In the START trial, which randomized PLHIV with CD4 > 500/mm^3^ to immediate cART initiation or cART deferral until CD4 dropped < 350/mm^3^, immediate cART initiation reduced the risk of all cancer by 64%. The risk of KS was also significantly reduced by 91% (95%CI; 0.01–0.99) but these estimates were based on a very low number of events (1 and 11 in the immediate and deferral group, respectively) and a relatively short follow-up of a median of 2.8 years (IQR 2.1–3.9). Recently, two large observational studies using causal inference methods have attempted to estimate the effect of early vs. deferred treatment on cancer risk on a larger population with a longer follow-up [54,55]. In the NA-ACCORD study [55], a protective effect of earlier ART for any virus-related cancer, mainly driven by the effect on KS, was observed (HR 0.25; 95%CI, 0.10–0.61). The DAD study [54] also estimated a protective effect of immediate cART initiation against AIDS-defining cancer, among which KS accounts for 2/3rd of cases, but to a lesser extent.

Overall, these results indicate that currently recommended earlier ART initiation may reduce the risk of developing KS. However, improvements in HIV diagnosis and in reducing the time to receive HIV care are needed to better measure the effect of earlier ART on KS risk.

### 3.5. Age and KS: Little Evidence for Premature Aging in PLHIV

In PLHIV, KS risk is slightly associated with higher age (see Figure 1) in Europe and North America and with a lower age in South Africa and Latin America [20].

In resource-rich settings, KS in PLHIV is generally diagnosed at an age of approximately 40 years for PLHIV and 60 years for the general population. The fact that KS apparently occurs at lower age in PLHIV than in the general population is largely due to the differences in the age structure of the two populations and not the premature aging induced by prolonged inflammation/activation that puts PLHIV at higher risk of age-related comorbidities [56]. Indeed, the proportion of PLHIV over 50 years of age is smaller among PLHIV than in the general population. Hleyhel et al. [22] used a methodology to correct for the difference in the age and sex structure of the HIV population relative to that of the general population [57]. They showed that although the mean age at KS diagnosis observed in the HIV population was 40.3 years and 57.5 years in the general population (observed difference of −17.2 years), the true difference was only −2.2 years after correction. Such a modest difference does not favor a role for premature aging in KS but rather earlier diagnosis of KS or earlier acquisition of HHV8 in PLHIV, resulting in the promotion of KS development in HIV-HHV8 co-infected patients.

### 3.6. Treatment of KS in PLHIV

In PLHIV, the advent of combined ART has not only dramatically reduced the risk of KS but also changed the management of KS. Indeed, cART is essential in managing KS when immunosuppression is reversible and is the first KS treatment option in PLHIV [58,59,60,61]. Most often, immune restoration leads by itself to KS lesions regression in several months in localized non-aggressive forms. In more aggressive form, systemic treatments are needed which may rely on chemotherapeutic agents such as liposomal doxorubicin or taxanes or immune-modulating therapy (interferon alpha or PEG-interferon), or antiangiogenic agents.

## 4. KS in Immune-Suppressed Transplanted Patients

Immunosuppressed organ-transplanted recipients have an increased risk of developing cancer associated with viruses, such as Epstein–Barr virus, HHV8, hepatitis B and C, and human papillomavirus [62,63,64,65,66,67]. The risk of KS is much higher in PLHIV than in transplanted recipients (relative risk 3624 vs. 208 relative to that of the general population, respectively) [64]. Post-transplant KS generally develops 2–3 years after transplantation [68,69,70] and is largely due to HHV8 reactivation in transplant receivers, although HHV8 can also be transmitted by donor organs [71,72].

Recently, a large American linkage study between a transplant registry and cancer registry [69] showed that the factors associated with post-transplant KS were also those associated with a higher prevalence of HHV8: male sex, nonwhite race, non-US citizenship, higher age at transplant, and lung transplant (vs kidney). In this study, from a country where the prevalence of HHV8 is low (<5%), the incidence rate after transplantation was 12.4/100,000 and significantly declined over time from 1987 to 2014. In another recent study reporting post-transplantation KS between 1997 and 2016 in Italy, where the prevalence of HHV8 is between 10 and 30%, the incidence rate was 10 times higher (123.7/100,000) and decreased over time [70]. Interestingly, the incidence rates were much lower for those receiving mTOR inhibitors than those who were not.

Post transplantation KS is generally managed by reducing the immunosuppressive treatment to the lowest levels compatible with allograft function or by changing the immunosuppressive agent, such as changing from calcineurin inhibitors to mTOR inhibitors [68,73].

There are currently no specific recommendations for HHV8 screening (as there is no specific treatment for HHV8 infection) nor for cancer or a fortiori KS screening in transplanted patients. Furthermore, there is a consensus that neither donor nor recipient anti-HHV8 antibodies preclude transplantation.

## 5. The Other Forms of KS: The Classic KS and Endemic KS

Apart from immunodeficiency induced by HIV/AIDS or by iatrogenic treatment, KS have drawn less attention in recent years. Classic KS occur mostly in elderly men living or emigrating from Eastern Europe and the Mediterranean Sea (Italy). Before the AIDS epidemic, classic KS incidence rate was estimated as 1.58/100,000 person-years in Sardinia [74] while it was only 0.014/100,000 person-years in the UK [75]. Endemic KS occurs in children or young adults living in regions where the HHV8 is endemic (sub-Saharan Africa). Some authors also described KS arising in MSM in absence of HIV infection as a fifth form of KS [58,73,76]. The underlying mechanisms for the development of these KS in addition to HHV8 infection are not well understood. They likely also involve a part of immune deficiency due to ageing for classic KS and to multiple chronic infections and malnutrition for endemic KS. Some environmental factors such as exposure to volcanic soils have been also hypothesized [77,78,79]. Chronic exposure of the skin to iron or alumino-silicate might induce localized immune dysfunction which might explain the topography of the lesions at the extremities of the body and its higher incidence in rural regions [77]. Additionally, some association with HLA [80] and other genetic factors with HHV8 infection susceptibility have been also postulated as predisposition factors for KS [81].

Classic KS are usually indolent and progress slowly while AIDS KS are more aggressive [47]. In a recent retrospective monocentric study of endemic and classic KS (29 and 131, respectively) in France, endemic KS was associated with higher risk of systemic treatment, chemotherapeutic agent or immune-modulating therapies, and initiation than classic KS [82]. There is no staging classification of classic, endemic, and post-transplant KS. [59]. Treatments either local or systemic are offered to achieve KS control and preserve the quality of life of the patients based on several factors including KS lesions presentation (the number and the topography of the lesions, the presence of symptomatic lesions, and aggressiveness) but also on patient’s fitness. New therapeutics for KS will be covered by another article of this special issue. Recently, some European guidelines have been published to help clinicians to drive the therapeutic choices according to the clinical form and extension of the KS [59].

## 6. Conclusions

KS is a relatively rare tumor that is still endemic in southern and east Africa and has been associated with immunosuppression. KS is generally manageable by the introduction of antiretroviral treatment therapy for PLHIV or by reducing the immunosuppressive treatment for immunosuppressed organ-transplanted recipients. Current challenges in resource-poor settings are early HIV diagnosis, early treatment, and access to treatment to lower the KS burden in these contexts.

## Figures and Tables

**Figure 1 cancers-13-05692-f001:**
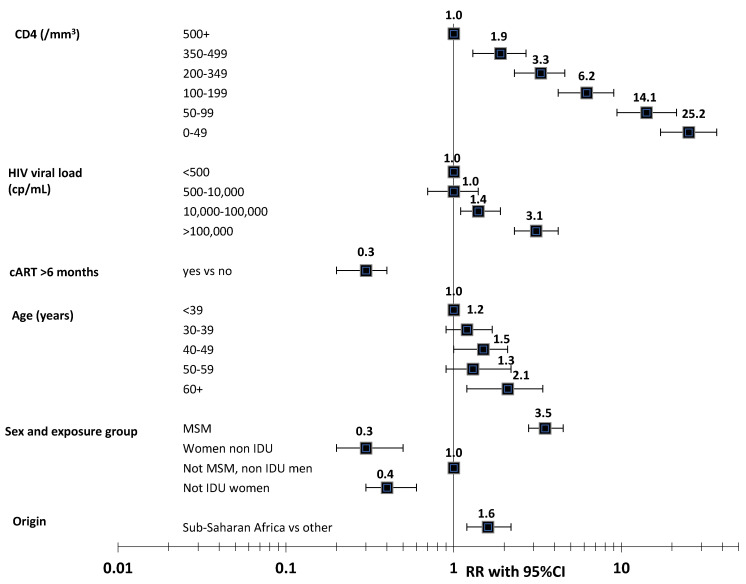
Risk factors of Kaposi’s sarcoma in PLHIV. Adapted from Guiguet et al., Lancet HIV 2009 [21]. Note: RR: relative risk; 95%CI: 95% confidence interval; MSM: men having sex with men; IDU: intravenous drug user. *p*-values: CD4: *p* < 10^−4^, HIV viral load: *p* < 10^−4^, exposure to cART: *p* < 10^−4^, age: *p* = 0.04, sex and exposure group: *p* < 0.0001, origin: *p* = 0.008.

**Figure 2 cancers-13-05692-f002:**
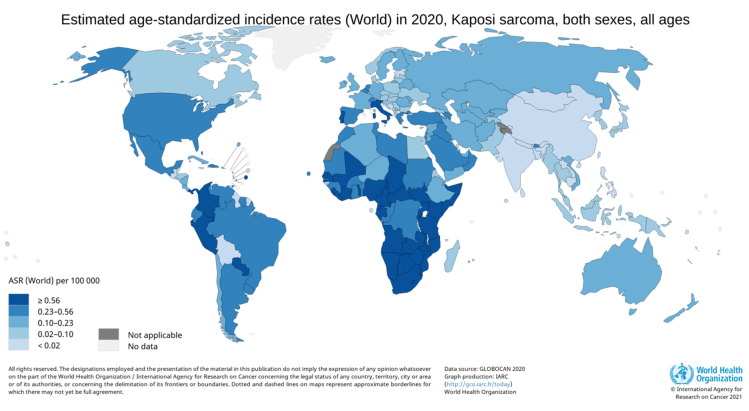
Incidence rates of Kaposi’s sarcoma in 2020. Data sources: Globocan 2020 (http://gco.iarc.fr/today, accessed on 29 June 2021).

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
