# Peer review of "Epidemiology of Kaposi’s Sarcoma"

_cancers, 2021, doi:10.3390/cancers13225692_

Round 1

Reviewer 1 Report

This resubmitted manuscript has substantially improved; however, not all the reviewer’s comments have been addressed adequately. There are still 2 main issues:

  1. The Simple Summary and the Abstract are still quite similar, and at least the (now deleted) sentence in the Abstract on what this review is about should go back in.
  2. KS/KSHV treatment: it is not enough to state in the text that treatment will be covered in another review. Specifically, comment 10 is about treatment for KSHV infection (not KS) and there is published evidence that some therapeutics (valganciclovir, foscarnet, cidofovir and rituximab) have promise in treatment of KSHV. Repurposing of therapeutics is a common form of drug discovery and should not just be disregarded because these drugs were not designed against KSHV. The addition of the sentence "new therapeutics for KS will be covered by another article of this special issue" has no relevance in response to this comment unless this other article is also going to cover KSHV treatments. Regarding comment 12, if therapeutics for KS will be covered in another review then it makes little sense to say "there are no curative treatments". If there are no curative treatments, then what is being covered in this other article of this special issue?

Author Response

Q1- The Simple Summary and the Abstract are still quite similar, and at least the (now deleted) sentence in the Abstract on what this review is about should go back in.

As suggested, the last sentence has been put back in the abstract of the revised manuscript.

Q2. KS/KSHV treatment: it is not enough to state in the text that treatment will be covered in another review. Specifically, comment 10 is about treatment for KSHV infection (not KS) and there is published evidence that some therapeutics (valganciclovir, foscarnet, cidofovir and rituximab) have promise in treatment of KSHV. Repurposing of therapeutics is a common form of drug discovery and should not just be disregarded because these drugs were not designed against KSHV. The addition of the sentence "new therapeutics for KS will be covered by another article of this special issue" has no relevance in response to this comment unless this other article is also going to cover KSHV treatments. Regarding comment 12, if therapeutics for KS will be covered in another review then it makes little sense to say "there are no curative treatments". If there are no curative treatments, then what is being covered in this other article of this special issue?

The special issue will also have an article specifically dedicated to KSHV. As suggested, we suppressed the sentence "there are no curative treatments" in the amended manuscript.

Reviewer 2 Report

The authors have described in this review the clinical aspect of KS notably in HIV-infected patients. The article summarized the KS variants and PLHIV treatments related to KS risk. The review is well structured and written.

In my opinion the article is suitable to publish in your journal. Minor revisions is required.

Minor points.

Q1. The authors must add a sub-section to describe the risk to develop or the progression of KS in PLHIV patients during targeted treatments. I would like to know the KS risk during treatments designed on HIV mutations by clinicians.

Author Response

Q1. The authors must add a sub-section to describe the risk to develop or the progression of KS in PLHIV patients during targeted treatments. I would like to know the KS risk during treatments designed on HIV mutations by clinicians.

As suggested, we have added a new paragraph on treatment in PLHIV in the revised manuscript.

"3.4 Treatments of KS in PLHIV:

In PLHIV, the advent of combined ART has not only dramatically reduced the risk of KS but also changed the management of KS. Indeed, cART is essential in managing KS when immunosuppression is reversible and is the first KS treatment option in PLHIV [58-61]. Most often, immune restoration leads by itself to KS lesions regression in several months in localized non-aggressive forms. In more aggressive form, systemic treatments are needed which may rely on chemotherapeutic agents such as liposomal doxorubicin or taxanes or immune-modulating therapy (interferon alpha or PEG-interferon), or antiangiogenic agent. "

This manuscript is a resubmission of an earlier submission. The following is a list of the peer review reports and author responses from that submission.

Round 1

Reviewer 1 Report

The submitted manuscript entitled "Epidemiology of Kaposi's Sarcoma" is a necessary contribution to the special issue "Perspectives on Kaposi's Sarcoma" in its aim to give an update on recent epidemiological evidence related to Kaposi's Sarcoma. However, care must be taken to differentiate this review from several other recent reviews on the topic, for example Cesarman et al. 2019 (DOI: s41572-019-0060-9), Vangipuram and Tyring 2019 (DOI: 10.1111/ijd.14080) and Dupin 2020 (DOI: 10.1097/CCO.0000000000000601). This manuscript can be improved to the level of publication, most importantly by including more up to date publications (currently 22/74 references date later than 2017). Below are some suggestions for improvement:

  1. The "simple summary" (line 10) and "abstract" (line 19) are identical except for a few words which does not come across well especially when reading the abstract directly after the simple summary as the manuscript will be laid out. This needs to be completely rewritten to differentiate the two sections.
  2. Error in keywords line 30: "KSVH" should be "KSHV"
  3. Line 17 (and line 28 - see comment 1): '"mainly focuses" is a redundant phrase.
  4. Line 52: There is discussion and evidence in the literature of including KS in MSM as a distinct fifth variant. This should be included here. 
  5. Line 69: reference 12 dates back to 2010. There are more recent reports of HHV-8(/KSHV) prevalence in South Africa. See Maskew et al. 2011 (DOI: 10.1186/1750-9378-6-22) and Blumenthal et al. 2019 (DOI: 10.1093/infdis/jiz180 )
  6. Line 83: reference 17 refers to GLOBOCAN 2018 data. GLOBOCAN 2020 has been released and should be used here (see Sung et al. 2021 (DOI: 10.3322/caac.21660))
  7. Figure 1: This figure graphically represents data presented in Guiguet et al. 2009 as is correctly referenced in the figure legend. While the presentation is appealing, this figure does not present any new information nor does it collate data from several studies. A much more informative figure would be one that is constructed from multiple studies reporting on risk ratios. For example, reference 18 reports much more recent data (2017) from several countries. Other studies reporting on Kaposi's Sarcoma risk could additionally be included in the figure to give a more meaningful presentation. 
  8. Line 92: it is not clear why GLOBOCAN 2012 data is being discussed here rather than more up to date GLOBOCAN 2020 data. The only reason for including previous GLOBOCAN data is if a comparison is to be made over time. If this is the intention here it needs to be more clearly communicated.
  9. Line 131: It is not clear if "2-3000" is a range or a typo
  10. Line 241: The statement made that "there is no specific treatment for HHV8 infection" is short sighted and inaccurate. It would provide a richer review to include studies showing promising results of treatments such as valganciclovir in combination with rituximab (Pantanowitz et al. 2008 DOI: 10.1186/1472-6890-8-7) or zidovudine (Uldrick et al. 2011 DOI: 10.1182/blood-2010-11-317610), ganciclovir (Casper et al. 2004 DOI: 10.1182/blood-2003-05-1721), cidofovir (Mazzi et al. 2001 DOI: 10.1097/00002030-200110190-00026) and foscarnet (Low et al. 1997 PMID: 9753702 and Luppi et al. 2002 DOI: 10.1097/00007890-200207150-00023). 
  11. Line 260: Reference 71 reports on genetic predisposition factors for all forms of KS (not only classic KS)
  12. Line 266: Again, there is plenty of literature regarding KS treatment that should be included in this discussion to make it more informative and nuanced rather than an inaccurate blanket statement that there are no curative treatments. Up to 80% of ART-naive patients with limited KS show regression of KS on HAART (Goncalves et al. 2017 DOI: 10.1097/QAD.0000000000001567;  Mosam et al. 2012 DOI: 10.1097/QAI.0b013e318251aedd; Bower et al. 2009 DOI: 10.1097/qad.0b013e32832d080d) while patients with extensive oral KS, gastrointestinal or non-nodal visceral KS or tumour-associated oedema or ulceration do not show good responses. Additionally, systemic chemotherapy (doxorubicin, vincristine, vinblastine and bleomycin) is used although 30% of patients do not respond (Mosam et al. 2012 as above). 
  13. Line 271: "SK" is a typo - assuming it is meant to be "KS"

Reviewer 2 Report

The authors have described in this review epidemiological studies about Kaposi’s Sarcoma disease and HIV infection. They have been highlighted the crosstalk between HIV and KSHV viruses in AIDS patients. The review has been well structured in many sections. In my opinion, the article is suitable for Cancers publication. However, the manuscript need minor revision.

Minor points.

Q1. The authors must describe “general population” using the words “worldwide populations”.

Q2. The authors must add section regarding KSHV molecular mechanism in KS pathogenesis. They add some references such as Mesri group studies about the crosstalk between KSHV latent/lytic infection and the onset of Kaposi’s Sarcoma. It is necessary to point out the cellular and viral proteins involved to highlight the new therapeutic strategies used to treat patients affected by this disease.

Q3.The authors must underline the mTOR inhibitors used in the treatments of KSHV -associated diseases. 

Q4. The authors point out the crosstalk between CD4 depletion and KSHV impaired infection. They must add some new references about it.

Q5. Do the authors any studies about autophagy and HIV- and KSHV- infections in Kaposi’s Sarcoma patients? It could be interesting to note the viral maturation and autophagy machinery to avoid immune system surveillance and sustain the viral replication, e.g. 

Q6. The authors must improve English.

Author Response

Please see the attachment for the point-by-point responses to Reviewer 2
